# Nourishing futures: Assessing nutritional health among children under-five years of age belonging to a particularly vulnerable tribal community in Southern India

Sneha Deepak Mallya[1⊙], Biju Soman[1⊙], Srikanth Gouspure[1], Pavan Kumar Thatoju[1], Dinesh M. Nayak[2], Ashwini Kumar[1], Harpreet Kaur[3], Ranjitha S. Shetty[1]*

1 Department of Community Medicine, Kasturba Medical College, Manipal Academy of Higher Education, Manipal, India, 2 Department of Pediatrics, Melaka Manipal Medical College (Manipal Campus), Manipal Academy of Higher Education, Manipal, India, 3 Division of Epidemiology and Communicable Diseases, Indian Council of Medical Research, Department of Health Research, Ministry of Health and Family Welfare, New Delhi, India

⊙ These authors contributed equally to this work.
* ranjitha.shetty@manipal.edu

## Abstract

Malnutrition is a major public health concern among tribal communities, where children face higher risk due to socio-economic disadvantages, poor healthcare access, and inadequate diets. Despite government programs, it persists, necessitating a focused assessment of its determinants. This study aimed to evaluate the nutritional status and its determinants among Koraga tribal children. A community-based cross-sectional study was conducted among children under-five years of age belonging to the Koraga community, a particularly vulnerable tribal group (PVTG) in southern Karnataka, India. Anthropometric measurements were taken, and data on socio-demographic characteristics, maternal and child health, and environmental conditions were collected using a structured proforma. Data analysis was performed using SPSS version 16.0 with logistic regression identifying factors associated with malnutrition. Among the 180 children included in the study, the median [interquartile range (IQR)] age of the children was 36 months (22.25, 46.00), and 52.2% were female. Low birth weight was reported in 40.2% of children, and 9.9% were currently anaemic. The study found a high prevalence of malnutrition among children under-five years of age, with 81.1% being stunted, 78.3% wasted, and 80.6% underweight. Multivariable analysis identified access to safe drinking water, maternal factors, nutritional factors, and dietary factors as significant contributors to undernutrition in this vulnerable population. The findings highlight the urgent need for targeted nutrition and health interventions in this PVTG. Strengthening maternal care, child feeding practices, sanitation, and healthcare access is essential to reduce malnutrition. A multi-sectoral approach is essential to break the cycle of undernutrition and improve overall child health outcomes.

**Data availability statement:** The dataset includes information collected from members of a Particularly Vulnerable Tribal Group (PVTG), which may contain sensitive community-level details. In accordance with the conditions of approval from the Institutional Ethics Committee of Kasturba Medical College and Kasturba Hospital, Manipal, the de-identified dataset cannot be made publicly available to protect participant confidentiality and community privacy. All data required to support the findings of this study are available within the manuscript. The minimal dataset necessary to replicate the study's results may be made available upon reasonable request to the corresponding author and is subject to review and approval by the Institutional Ethics Committee. Requests may be directed to the institutional non-author data access contact at communitymed@manipal.edu. (Phone:+91-820-2922324).

**Funding:** Initial of the author who received the award: RSS; Grant number: (Seed money reference ID 00000525); Name of funder: Manipal Academy of Higher Education, Manipal. Funders did not play any role in study design, data collection and analysis, decision to publish and preparation of manuscript.

**Competing interests:** The authors have declared that no competing interests exist.

## Introduction

Malnutrition is a vicious cycle in both the general population and tribal communities, but it is often more severe among tribal populations due to greater socio-economic disadvantages, traditional practices, geographical isolation, limited health services, cultural beliefs, and lower literacy rates exacerbating the cycle. Malnutrition, as classified by the World Health Organization (WHO), includes both undernutrition (wasting, stunting, underweight, and micronutrient deficiencies) and overnutrition (overweight and obesity). Wasting (low weight for height/length) indicates acute malnutrition, while stunting (low height/length for age) reflects chronic undernutrition and underweight (low weight for age) can result from both, affecting growth and survival [1]. In 2022, 149 million children under-five years of age were stunted, 45 million were wasted, and 37 million were overweight or obese. Nearly half of deaths among children who are under-five years of age are linked to undernutrition, with lasting impacts in low- and middle-income countries [2,3]. Malnutrition, low birth weight (LBW), and diarrhea are major health problems among the tribal population of India. Many tribal individuals initially seek care from traditional healers and turn to public health facilities only when necessary. As a result, tribal children suffer from higher levels of malnutrition and increased rates of illness compared to their non-tribal counterparts [4–7].

The National Nutrition Monitoring Bureau (NNMB) 3rd Repeat Survey-2012 and Comprehensive National Nutrition Survey (CNNS) 2016–18 from India have shown a high burden of malnutrition among Scheduled Tribe (ST) children. According to the CNNS report, 41.5% of ST children under the age of five years were stunted, 21.9% were wasted, and 41.5% were underweight, indicating severe nutritional challenges [8]. Data from NNMB shows that ST children had the highest rates of underweight (54%), stunting (54%), and wasting (29%). These findings highlight severe health inequalities, caused by poverty, poor diet, lack of healthcare, and socio-cultural factors [9]. Malnutrition among ST children remains higher than in the general population. According to the National Family and Health Survey-5 (NFHS-5) of India, 40.9% of ST children are stunted, 23.2% are wasted, and 39.5% are underweight, compared to 35.5%, 19.3%, and 32.1% in the general population, respectively [10]. Malnutrition is also highly prevalent among tribal children under-five years of age in Karnataka, as indicated by stunting at 39.5%, wasting at 21.5%, and underweight at 35.8%. Additionally, 69.6% of children under-five years of age have anaemia. Further, the district of Udupi, where the present study was conducted, reported lower rates as compared to the state and national average with stunting at 23.1%, wasting at 17.6%, and underweight at 21.0% among children under-five years of age [11]. Of the 75 Particularly vulnerable tribal groups (PVTG) identified by the government of India, the two tribes namely Koraga and Jenukuruba from Karnataka, have poorer health status than other communities, due to poverty, poor sanitation, limited healthcare access, and their remote locations [12–14].

Tribal populations in India face multiple health challenges, including malnutrition, communicable and non-communicable diseases (NCDs), and mental health disorders. Traditional beliefs, substance abuse, and various socio-economic and cultural factors contribute to poor health outcomes, resulting in higher malnutrition rates

and limited access to healthcare. Among tribal children, socioeconomic status, indigenous background, parental factors, child-related factors, poor hygiene, and child illnesses are key contributors to stunting, wasting, and underweight, increasing the risk of morbidity and mortality [15,16]. PVTGs have poorer health indicators compared to both the general and other tribal populations. Limited access to maternal and child healthcare, poor health-seeking behavior, and high rates of substance abuse contribute to their vulnerability to malnutrition, anemia, and infectious diseases [17]. The Koraga community, a PVTG in coastal Karnataka, remains socially marginalized with limited access to health care due to geographical isolation, social stigma, language barriers and socioeconomic challenges. Prior findings from Koraga community indicated that 42.3% of under 5 children were underweight, 30.8% were stunted and 15.4% were wasted [12,18]. Addressing these issues aligns with Sustainable Development Goal (SDG) 2 (Zero Hunger) which emphasizes improving nutrition and SDG 3, which aims to ensure healthy lives and well-being for all [19]. Therefore, this study aimed to assess the nutritional status and its determinants among children under-five years of age belonging to the Koraga community, a PVTG residing in the Udupi district.

## Materials and methods

The study was approved by the Ethics Committee of Kasturba Medical College and Kasturba Hospital (IEC:747/2021). The data was collected from 3/12/2021–31/10/2022. After explaining the details about the study in the local language Kannada, a written informed consent was obtained from the mothers of children under-five years of age. For mothers who were illiterate a thumb impression was taken in the presence of a witness along with their signature who could read, write and understand the study related information. Community engagement was facilitated through local leaders and health workers.

Additional information regarding the ethical, cultural, and scientific considerations specific to inclusivity in global research is included in the supporting information (S1 File).

This community-based cross-sectional study was conducted among the children under-five years of age belonging to Koraga tribe residing in Udupi district. The district of Udupi is located in Karnataka, a south Indian state. The Koraga population resides across seven blocks in Udupi District including Kundapura, Kaup, Karkala, Byndoor, Udupi, Hebri and Brahmavara. A complete enumeration sampling approach was adopted. All the tribal hamlets were visited, and eligible children were included. After complete enumeration of all eligible children in the study setting, a total of 180 children constituted for the final sample size of the study. The list of eligible houses of the Koraga community was obtained prior to initiation of the survey from Integrated Tribal Development Project (ITDP) Office, Udupi, an administrative body by Government of India to support the socio-economic development of scheduled tribes. Following this, door-to-door visits were conducted with the help of local community leaders and the purpose of the visit was explained to the mothers of children within the family. All the eligible children in a given household who were aged ≤ five years were recruited into the study. Socio-demographic details of the household were collected by administering a pre-tested, semi-structured questionnaire for the mothers of children with the assistance of a qualified nurse serving as a research assistant. Socio-economic status was assessed by using modified BG Prasad's scale [20]. Details regarding prenatal, natal and post-natal history, breastfeeding, feeding practices, immunization, and any recent infections were collected. Details pertaining to utilization of nutritional supplementation services provided through Integrated Child Development Services (ICDS) scheme and rations obtained from ITDP were collected. The ICDS program, is a flagship initiative of the Government of India, that provides essential nutrition, healthcare, and early education to children under six years old, significantly improving their health and development outcomes through Anganwadi centers [21]. Accredited social health activist (ASHA) at gross root level supports Anganwadi centers by mobilizing pregnant, lactating women and infant/children for nutritional supplementation and related health services through regular home visits. The food packets provided by ITDP include essential items such as rice, pulses, oil, and other basic food supplies to ensure balanced nutrition. A detailed physical examination was carried out by the study nurse among all the recruited children to look for signs of nutritional deficiency, which included the

presence of pallor, cheilosis/angular stomatitis (Vitamin B2 deficiency) and Bitot's spots (Vitamin A deficiency). History of acute respiratory infections and acute diarrheal disorders, chronic ingestion of inedible substances (pica) and worm infestation were enquired about. Nutritional intake of mothers and children was assessed by using the 24 hours dietary recall method with respect to calories, protein, vitamin A and iron intake as per National Institute of Nutrition (NIN) guidelines [22]. Length and weight for age and weight for length were assessed for children aged 0–24 months; height and weight for age and weight for height were assessed for children aged 24–59 months using the WHO 2006 guidelines. Stunting, wasting and underweight are key anthropometric indicators of child malnutrition. Children with height for age, weight for height or weight for age, Z scores below −2 standard deviations are classified as stunted, wasted or underweight respectively [23]. The mothers' height and weight were measured and categorized based on the body mass index (BMI) values as per WHO classification [24]. Screening for anemia was carried out through finger prick method using a handheld portable hemoglobinometer for both mothers and children (1–5 years) at the household level and was classified according to WHO guidelines [25]. Peripheral smears were done using finger prick method for those children who were found to be anemic and mothers who had moderate and severe anemia and the slides were tested at Department of Pathology of the medical college. Data was entered and analyzed using SPSS version 16.0. Results have been expressed as frequencies and proportions. Continuous variables are presented as mean or median with the standard deviation (SD) or interquartile range (IQR) respectively wherever applicable. Variables with a p-value ≤0.2 on univariate analysis were included for multi-variable analysis. Multivariable logistic regression analysis was done to find out the association between stunting, wasting and underweight and various background characteristics, including maternal and child-related parameters. The strength of association has been expressed as unadjusted odd's ratio (OR) and adjusted OR (AOR) with 95 per cent confidence interval (CI). A p-value <0.05 was considered statistically significant.

## Results

The study included a total of 180 households and a third of the households were from Kundapura block. The proportion of households included from other six blocks were as follows: Kaup (14.4%), Karkala (14.4%), Byndoor (14.4%), Udupi (11.2%), Hebri (8.3%), Brahmavara (5.6%). The background characteristics of the households included in the study are depicted in Table 1. Among the study participants, 91.1% possessed their own houses, though 133 (73.9%) lived in semi-pucca houses. Of the households surveyed, 168 (93.3%) reported having an electrical connection, and only 37.2% of the families used LPG as cooking fuel. Almost one-fourth (23.3%) of them did not have a habit of treating water before consuming, and 7.2% of the families did not have a household toilet. Further, more than two-thirds (69.4%) of the families possessed a below poverty line (BPL) ration card, and 171 (95.0%) reported having received ITDP food packets.

Table 2 depicts reproductive characteristics, dietary intake and nutritional profile of surveyed mothers. The majority (74.4%) of mothers had completed 5–10 years of schooling. The mean (SD) age at marriage and first conception was 22.72(4.66) years and 24.07(4.83) years, respectively. Among the study participants, 18(10.0%) had conceived more than three times.

While querying about dietary intake, it was found that the calorie and calcium intake were inadequate among all mothers. On anthropometric assessment, almost half (45.0%) of the mothers were found to be underweight, and 43.9% were found to have anaemia on hemoglobin estimation. Peripheral smear examination was conducted among 30 mothers diagnosed with moderate to severe anaemia. Of these, two were found to have microcytic anaemia, while the remaining 28 exhibited normocytic anaemia. Peripheral smear could not be performed on seven mothers.

Mothers of children under-five years of age were asked about their source of information regarding the services provided by the Anganwadi (a center under ICDS for delivering the services related to childcare and nutrition). The responses included their mothers (148, 82.3%), health workers (25,13.9%) and other relatives (7,3.9%). The participants were aware of the following services: nutritious food (123, 68.3%), non-formal pre-school education (49, 27.2%) and vaccination (5, 2.8%). About three mothers (1.7%) were not aware of any services provided by the Anganwadi.

 

**Table 1. Background characteristics of the surveyed households (n = 180).**

| Characteristics | Categories | n (%) |
| --- | --- | --- |
| Type of family | Nuclear | 48 (26.7) |
| | Joint | 112 (62.2) |
| | Third generation | 20 (11.1) |
| Family members | ≤ 5 members | 69(37.8) |
| | >6 members | 112 (62.2) |
| Socioeconomic status | Upper middle (3766–7532) | 9 (5.0) |
| | Middle (2260–3765) | 41 (22.8) |
| | Lower middle (1130–2259) | 72 (40.0) |
| | Lower (≤1129) | 58 (32.2) |
| Possession of Ration Card | Yes | 170 (94.4) |
| | No | 10 (5.6) |
| Type of house | Kutcha House | 34 (18.9) |
| | Semi-Pucca house | 133 (73.9) |
| | Pucca house | 13 (7.2) |
| Type of cooking Fuel | Wood stove | 113 (62.8) |
| | Liquid Petroleum Gas | 67 (37.2) |
| Source of drinking water | Open well | 96 (53.3) |
| | Bore well | 84 (46.7) |
| Drinking water treatment before use | Yes | 138 (76.7) |
| | No | 42 (23.3) |
| Presence of a household toilet | Yes | 167 (92.8) |
| | No | 13 (7.2) |

Characteristics of children, including details on breastfeeding, current nutritional intake, anthropometric parameters and morbidity are depicted in Table 3. The median (IQR) age of the children was 36.00 (22.25, 46.00) months, 52.2% were female and 40.2% of children were born with low birth weight. Of the surveyed children, 145(80.6%) were underweight, 146 (81.1%) were stunted and 141(78.3%) were wasted. Hemoglobin was estimated among 161 children and was found to be normal among 145 (90.1%). The mean (SD) hemoglobin level was 10.93(3.37) gm/dl. Peripheral smear examination was performed on 13 out of 16 children diagnosed with anaemia and three children exhibited microcytic anaemia, while the remaining ten showed normocytic anaemia. Calorie, calcium and iron intake were inadequate among the majority of children.

Table 4 depicts univariate and multivariable logistic regression analysis between stunting and sociodemographic, maternal and child characteristics.

Univariate analysis showed an association of stunting with source of drinking water, treatment of drinking water before use, maternal retinol intake, low birth weight, worm infestation, retinol and iron intake of the children. Further, on multivariable analysis, the source of drinking water (AOR: 4.02; CI: 1.08–14.89; p = 0.03), a higher number of maternal pregnancies (AOR: 4.47; CI: 1.03–19.31; p = 0.04), low birth weight (AOR: 6.69; CI: 1.68–26.59; p = 0.007) and retinol inadequacy (AOR: 6.31; CI: 1.54–25.83; p = 0.01) were positively associated with stunting. However, lack of exclusive breastfeeding for the first six months, inadequate home visits by ASHA/CHO and worm infestation in the children showed a negative association with stunting. These negative associations may be influenced by the high prevalence of stunting within the study population or the limited sample size in certain categories of these variables.

Similarly, for wasting among children under-five years of age, univariate and multivariable analyses showed significant associations with age of the child and poor dietary protein intake as depicted in Table 5. The present study results indicate

**Table 2. Reproductive characteristics, dietary intake and nutritional profile of the surveyed mothers (n = 178).**

| Characteristics | Categories | n (%) |
|---|---|---|
| Educational status | Illiterate up to 4 years of schooling | 25 (13.9) |
| | 5-10 years of schooling | 134 (74.4) |
| | >10 years of schooling | 21(11.7) |
| Age at marriage (in years) | 15-17 | 19(10.6) |
| | ≥18-25 | 161(89.4) |
| Age at first conception (in years) | <18 | 5(2.8) |
| | 18-30 | 153(85.0) |
| | >30 | 22(12.2) |
| Number of times conceived | Once | 90 (50.0) |
| | Twice | 72 (40.0) |
| | Thrice | 12 (6.7) |
| | >than 3 times | 6 (3.3) |
| Type of delivery | Normal | 99 (55.0) |
| | Cesarean section | 81 (45.0) |
| Calorie intake of mother | Inadequate | 178 (100.0) |
| Protein intake of mother | Adequate | 8 (4.5) |
| | Inadequate | 170 (95.5) |
| Calcium intake of mother | Inadequate | 178 (100.0) |
| Iron intake of mother | Adequate | 3 (1.7) |
| | Inadequate | 175 (98.3) |
| Retinol intake of mother | Adequate | 20 (11.2) |
| | Inadequate | 158 (88.8) |
| BMI of mother | Under weight | 81 (45.0) |
| | Normal weight | 77 (42.8) |
| | Overweight &Obesity | 22 (12.2) |
| Hemoglobin level of mother | Normal | 100 (56.1) |
| | Mild anaemia | 41 (23.1) |
| | Moderate anaemia | 14(7.9) |
| | Severe anaemia | 23 (12.9) |

Note: One mother had twins and another mother had two children under five years, resulting in a total of 178 mothers.

that children aged 25−60 months had a higher likelihood of wasting (AOR: 5.05; CI: 2.13–11.99; p < 0.001) while inadequate protein intake was associated with a lower likelihood of wasting (AOR: 0.27; CI: 0.11–0.69; p = 0.006). This may be attributed to the fact that most of the children were wasted.

As shown in Table 6, age ≤ 30 years at conception was four times more likely to be associated with underweight, however, this was not statistically significant (AOR: 4.47; 0.98–20.26; p = 0.05). Due to the very high proportion of underweight children in the study population, not having a household toilet and not preparing fresh meals showed a negative association with underweight.

## Discussion

The present study found that malnutrition was highly prevalent among Koraga tribal children, with 81.1% of children being stunted, 78.3% of the children being wasted, and 80.6% of the children being underweight. Malnutrition rates reported

**Table 3. Anthropometry, dietary and morbidity profile of children under-five years of age (n=180).**

| Characteristics | Categories | n (%) |
|---|---|---|
| Age (in months) | 0-12 | 11 (6.1) |
| | 13-24 | 47 (26.1) |
| | 25-60 | 122 (67.8) |
| Sex | Male | 86 (47.8) |
| | Female | 94 (52.2) |
| Hospitalization during neonatal period | Yes | 21 (11.7) |
| | No | 159(88.3) |
| Birth weight | Very low birth weight | 10 (5.6) |
| | Low birth weight | 62 (34.6) |
| | Normal weight | 107 (59.8) |
| Weight for age | Severe underweight | 21 (11.7) |
| | Moderate underweight | 75 (41.7) |
| | Mild underweight | 49 (27.2) |
| | Normal weight | 35 (19.4) |
| Height for age | Severe Stunting | 23 (12.8) |
| | Moderate Stunting | 35 (19.4) |
| | Mild Stunting | 88 (48.9) |
| | normal status | 34 (18.9) |
| Weight for height | Severe wasting | 46 (25.6) |
| | Moderate wasting | 34 (18.9) |
| | Mild wasting | 61 (33.9) |
| | Normal status | 39 (21.7) |
| Hemoglobin level | Normal | 145 (90.1) |
| | Mild anaemia | 1(0.6) |
| | Moderate anaemia | 10 (6.2) |
| | Severe anaemia | 5 (3.1) |
| Type of anaemia | Normocytic anemia | 10 (5.5) |
| | Microcytic anemia | 3 (1.7) |
| Immunization status | Fully immunized | 176 (97.8) |
| | Partially immunized | 4 (2.2) |
| History of Measles in the last six months | Yes | 9 (5.0) |
| | No | 171 (95.0) |
| History of Acute Respiratory Infection/Acute Diarrheal Disease in the last six months | Yes | 11(6.1) |
| | No | 169(93.9) |
| Worm infestation in last six months | Yes | 15 (8.3) |
| | No | 165 (91.7) |
| History of Ingestion of clay/soil | Yes | 18 (10.0) |
| | No | 162 (90.0) |
| Hospitalization of child in the last year for acute illness | Yes | 36 (20.0) |
| | No | 144 (80.0) |
| Calorie intake | Adequate | 34 (18.9) |
| | Inadequate | 146 (81.1) |
| Protein intake | Adequate | 119 (66.1) |
| | Inadequate | 61 (33.9) |

*(Continued)*

**Table 3.** (Continued)

| Characteristics | Categories | n (%) |
|---|---|---|
| Calcium intake | Adequate | 7 (3.9) |
| | Inadequate | 173 (96.1) |
| Iron intake | Adequate | 37 (20.6) |
| | Inadequate | 143 (79.4) |
| Retinol intake | Adequate | 71 (39.4) |
| | Inadequate | 109(60.6) |
| Breast feeding initiation | Immediately after birth | 107 (59.4) |
| | Half an hour-4 hours | 68 (37.7) |
| | Beyond 4 hours | 5 (2.7) |
| Pre-lacteal feeding to the infant | Yes | 21 (11.7) |
| | No | 159 (88.3) |
| Duration of breastfeeding to child (in months) | 0 - 6 | 10 (5.6) |
| | 7 −12 | 12 (6.7) |
| | 13 −24 | 60 (33.3) |
| | > 24 | 98 (54.4) |
| Whether mother breastfeeds when she is sick | Yes | 134 (74.4) |
| | No | 46 (25.6) |
| Whether mother breastfeeds when the child is sick | Yes | 137 (76.1) |
| | No | 43 (23.9) |
| Whether washes hands before feeding the child | Yes | 118 (65.6) |
| | No | 10 (5.6) |
| | Sometimes | 52 (28.8) |
| Washes fruits and vegetables | Yes | 176 (97.7) |
| | No | 4 (2.2) |
| Freshly prepares meals | Yes | 122 (67.8) |
| | No | 58 (32.2) |
| Supplementary nutrition provided by ICDS is utilized by child | Yes | 153 (85.0) |
| | No | 23 (12.8) |
| | Not sure | 4 (2.2) |
| Whether house is visited by accredited social health activist (ASHA)/ Community health officer (CHO)[a] | Yes | 165(91.7) |
| | No | 15(8.3) |
| Presence of nearby health facility | Yes | 175(97.2) |
| | No | 5(2.8) |

Note: Nine children were not tested for anaemia since they were below one year, 10 children refused haemoglobin test.

(CHO)[a] – grassroot level workers who are responsible for house visit at the subcenter or primary health center level.

in different studies across various states in India range from 8.5% to 75.26% [13,26–32]. According to NFHS-5, stunting, wasting, and underweight are found to be 39.5%, 21.5%, and 35.8% respectively among ST children in Karnataka. However, the overall (non-tribal and tribal children) prevalence of stunting, wasting, and underweight in Udupi district were at 23.1%, 17.6%, and 21.0%, respectively, which is significantly lower than our study findings. The lower rates in Udupi district may be due to the unavailability of ST-specific nutritional data from NFHS-5 [10].

The prevalence rates of malnutrition are higher among children from PVTGs. A study among children belonging to PVTGs of Odisha by Das A et al. found that 75.26% were underweight, 55.42% were stunted, and 60% were wasted [13],

**Table 4. Association between stunting and sociodemographic, maternal and child characteristics (n = 180).**

| Characteristics | Category | Stunting | | COR (95%CI) | p value | AOR (95%CI) | p value |
|---|---|---|---|---|---|---|---|
| | | Present (n = 146) n(%) | Absent (n = 34) n(%) | | | | |
| **Sociodemographic characteristics** | | | | | | | |
| Type of family | Nuclear | 44(91.7) | 4(8.3) | 1.22 (0.02-7.27) | 0.82 | 1.52 (0.55-42.13) | 0.803 |
| | Joint | 84(75.0) | 28(25.0) | 0.33 (0.07-1.52 | 0.15 | 0.17 (0.01-1.809) | 0.14 |
| | Third generation | 18(90.0) | 2(10.0) | 1 | | 1 | |
| Number of Family members | ≤5 members | 59(86.8) | 9(13.2) | 1 | | 1 | |
| | >6 members | 87(77.7) | 25(22.3) | 0.053 (0.23-1.21) | 0.13 | 0.28 (0.04-1.98) | 0.206 |
| Socioeconomic status | Middle | 102(83.6) | 20(16.4) | 1 | | 1 | |
| | Lower | 44(75.9) | 14(24.1) | 0.61 (0.28-1.33) | 0.21 | 0.69 (0.17-2.69) | 0.59 |
| Type of ration card | APL | 10(90.9) | 1(9.1) | 1 | | – | |
| | BPL/ Antyodaya/No ration card | 136(80.5) | 33(19.5) | 0.41 (0.05-3.33) | 0.406 | – | |
| Type of house | Kutcha House | 25(73.5) | 9(26.5) | 0.23 (0.02-2.04) | 0.18 | 0.05 (0.003-1.209) | 0.66 |
| | Semi-Pucca house | 109(82.0) | 24(18.0) | 0.37 (0.04-3.05) | 0.36 | 0.33 (0.02-5.54) | 0.44 |
| | Pucca house | 12(92.3) | 1(7.7) | 1 | | 1 | |
| Type of cooking fuel | Wood stove | 94(83.2) | 19(16.8) | 1.42 (0.66-3.04) | 0.35 | – | |
| | LPG stove | 52(77.6) | 15(22.4) | 1 | | – | |
| Source of drinking water | Open well | 83(86.5) | 13(13.5) | 2.12 (0.99-4.57) | 0.05 | 4.02 (1.08-14.89) | 0.03* |
| | Bore well | 63(75.0) | 21(25.0) | 1 | | 1 | |
| Drinking water treatment before use | Yes | 107(77.5) | 31(22.5) | 1 | | 1 | |
| | No | 39(92.9) | 3(7.1) | 3.76 (1.08-13.02 | 0.03 | 7.26 (0.80-65.73) | 0.07 |
| Presence of household toilet | Yes | 135(80.8) | 32(19.2) | 1 | | – | |
| | No | 11(84.6) | 2(15.4) | 1.30 (0.27-6.17) | 0.73 | – | |
| **Maternal characteristics** | | | | | | | |
| Educational status of mother | Illiterate or upto 4 years of schooling | 20(80.0) | 5(20.0) | 1.25 (0.307-5.08) | 0.75 | – | |
| | 5-10 years of schooling | 110(82.1) | 24(17.9) | 1.43 (0.47-4.29) | 0.52 | – | |
| | >10 years of schooling | 16(76.2) | 5(23.8) | 1 | | – | |
| Education of spouse | Illiterate or upto 4 years of schooling | 51(83.6) | 10(16.4) | 1.02 (0.28-3.63) | 0.97 | – | |
| | 5-10 years of schooling | 75(78.9) | 20(21.1) | 0.75 (0.23-2.44) | 0.63 | – | |
| | >10 years of schooling | 20(83.3) | 4(16.7) | 1 | | – | |
| Age at marriage (in years) | <18 | 14(73.7) | 5(26.3) | 0.61 (0.205-1.84) | 0.38 | – | |
| | ≥18 | 132(82.0) | 29(18.0) | 1 | | – | |

*(Continued)*

| Characteristics | Category | Stunting | | COR (95%CI) | p value | AOR (95%CI) | p value |
|---|---|---|---|---|---|---|---|
| | | Present (n=146) n(%) | Absent (n=34) n(%) | | | | |
| Age at first conception (in years) | ≤30 | 129(81.6) | 29(18.4) | 1.308 (0.446-3.83) | 0.62 | – | |
| | >30 | 17(77.3) | 5(22.7) | 1 | | – | |
| Number of times conceived | Once | 69(76.7) | 21(23.3) | 1 | | 1 | |
| | ≥ Two | 77(85.6) | 13(14.4) | 1.803 (0.84-3.87) | 0.13 | 4.47 (1.03-19.31) | 0.04* |
| Number of ANC check ups | <Four visits | 9(90.0) | 1(10.0) | 2.16 (0.26-17.71) | 0.47 | – | |
| | ≥Four visits | 137(80.6) | 33(19.4) | 1 | | – | |
| Type of delivery | Normal | 77(77.8) | 22(22.2) | 1 | | 1 | |
| | Cesarean section | 69(85.2) | 12(14.8) | 1.64 (0.75-3.56) | 0.209 | 2.83 (0.76-10.58) | 0.12 |
| Calorie intake among women | Inadequate | 34(19.1) | 144(80.9) | – | | – | |
| Protein intake among women | Adequate | 6(75.0) | 2(25.0) | 1 | | – | |
| | Inadequate | 138(81.2) | 32(18.8) | 1.43 (0.27-7.45) | 0.66 | – | |
| Calcium intake among women | Inadequate | 144(80.9) | 34(19.1) | – | | – | |
| Iron intake among women | Adequate | 2(66.7) | 1(33.3) | 1 | | – | |
| | Inadequate | 142(81.1) | 33(18.9) | 2.15 (0.18-24.44) | 0.53 | – | |
| Retinol intake among women | Adequate | 12(60.0) | 8(40.0) | 1 | | – | |
| | Inadequate | 132(83.5) | 26(16.5) | 3.38 (1.26-9.09) | 0.016 | 4.81 (0.706-32.807) | 0.108 |
| BMI of mother | Under weight | 65(80.2) | 16(19.8) | 1.15 (0.53-2.48) | 0.71 | 0.22 (0.04-1.03) | 0.05 |
| | Normal weight | 60(77.9) | 17(22.1) | 1 | | 1 | |
| | Overweight & Obesity | 21(95.5) | 1(4.5) | 5.95 (0.74-47.48) | 0.09 | 28.50 (0.32-2535.66) | 0.14 |
| Hemoglobin level of mother | Normal | 81(81.0) | 19(19.0) | 1 | | – | |
| | Mild anaemia | 33(80.5) | 8(19.5) | 1.56 (0.43-5.80) | 0.504 | – | |
| | Moderate anaemia | 10(71.4) | 4(28.6) | 0.58 (0.16-2.07) | 0.407 | – | |
| | Severe anaemia | 20(87.0) | 3(13.0) | 0.96 (0.38-2.42) | 0.94 | – | |
| **Characteristics of children** | | | | | | | |
| Age (in months) | Below 24 months | 50(86.2) | 8(13.8) | 1 | | 1 | |
| | 25-60 months | 96(78.7) | 26(21.3) | 0.59 (0.24-1.40) | 0.23 | 0.34 (0.08-1.45) | 0.14 |
| Sex | Male | 71(82.6) | 15(17.4) | 1 | | – | |
| | Female | 75(79.8) | 19(20.2) | 0.83 (0.39-1.76) | 0.63 | – | |
| Hospital admission in neonatal period | Yes | 18(85.7) | 3(14.3) | 1.45 (0.403-5.24) | 0.56 | – | |
| | No | 128(80.5) | 31(19.5) | 1 | | – | |

*(Continued)*

| Characteristics | Category | Stunting | | COR (95%CI) | p value | AOR (95%CI) | p value |
|---|---|---|---|---|---|---|---|
| | | Present (n = 146) n(%) | Absent (n = 34) n(%) | | | | |
| Birth weight | Low birth weight | 63(87.5) | 9(12.5) | 2.13 (0.93-4.89) | 0.07 | 6.69 (1.68-26.59) | 0.007* |
| | Normal weight | 82(76.6) | 25(23.4) | 1 | | 1 | |
| Immunization status | Fully immunized | 145(82.4) | 31(17.6) | 1 | | 1 | |
| | Partially immunized | 1(25.0) | 3(75.0) | 0.07 (0.007-0.708) | 0.024 | 0.11 (0.003-4.38) | 0.24 |
| Positive history of Measles in the last six months | Yes | 6(66.7) | 3(33.3) | 0.44 (0.105-1.86) | 0.26 | – | |
| | No | 140(81.9) | 31(18.1) | 1 | | – | |
| History of ARI/ADD in last six months | Yes | 8(72.7) | 3(27.3) | 0.59 (0.15-2.38) | 0.46 | – | |
| | No | 138(81.7) | 31(18.3) | 1 | | – | |
| Worm infestation in last six months | Yes | 9(60.0) | 6(40.0) | 0.30 (0.10-0.93) | 0.03 | 0.05 (0.006-0.47) | 0.008* |
| | No | 137(83.0) | 28(17.0) | 1 | | 1 | |
| History of Ingestion of clay/soil | Yes | 14(77.8) | 4(22.2) | 0.79 (0.24-2.58) | 0.704 | – | |
| | No | 132(81.) | 30(18.5) | 1 | | – | |
| Hospitalization of child in last one year for acute illness | Yes | 27(75.0) | 9(25.0) | 0.63 (0.26-1.503) | 0.29 | – | |
| | No | 119(82.6) | 25(17.4) | 1 | | – | |
| Calorie intake | Adequate | 24(70.6) | 10(29.4) | 1 | | 1 | |
| | Inadequate | 122(83.6) | 24(16.4) | 2.11 (0.89-4.99) | 0.08 | 0.49 (0.08-2.75) | 0.42 |
| Protein intake | Adequate | 94(79.0) | 25(21.0) | 1 | | – | |
| | Inadequate | 52(85.2) | 52(85.2) | 1.52 (0.66-3.53) | 0.31 | – | |
| Calcium intake | Adequate | 5(71.4) | 2(28.6) | 1 | | – | |
| | Inadequate | 141(81.5) | 32(18.5) | 1.76(0.32-9.49) | 0.51 | – | |
| Iron intake | Adequate | 25(67.6) | 12(32.4) | 1 | | 1 | |
| | Inadequate | 121(84.6) | 22(15.4) | 2.64 (1.15-6.02) | 0.02 | 2.11 (0.45-9.82) | 0.34 |
| Retinol intake | Adequate | 50(70.4) | 21(29.6) | 1 | | 1 | |
| | Inadequate | 96(88.1) | 13(11.9) | 3.10 (1.43-6.709) | 0.004 | 6.31 (1.54-25.83) | 0.01* |
| Breast feeding initiation | Immediately after birth | 87(81.3) | 20(18.7) | 1 | | – | |
| | Half an hour-4 hours | 55(80.9) | 13(19.1) | 0.97 (0.44-2.11) | 0.94 | – | |
| | Beyond 4hrs | 4(80.0) | 1(20.0) | 0.92 (0.09-8.67) | 0.94 | – | |
| Pre lacteal feeding to infant | Yes | 14(66.7) | 7(33.3) | 0.409 (0.15-1.109) | 0.07 | 2.98 (0.58-15.209) | 0.18 |
| | No | 132(83.0) | 27(17.0) | 1 | | 1 | |

*(Continued)*

**Table 4.** (Continued)

| Characteristics | Category | Stunting | | COR (95%CI) | p value | AOR (95%CI) | p value |
|---|---|---|---|---|---|---|---|
| | | Present (n = 146) n(%) | Absent (n = 34) n(%) | | | | |
| Duration of breast feeding to child (months) | Upto 6 | 6(60.0) | 4(40.0) | 0.36 (0.09-1.407) | 0.14 | 0.04 (0.005-0.39) | 0.005* |
| | 7 −12 | 8(66.7) | 4(33.3) | 0.48 (0.13-1.76) | 0.27 | 0.11 (0.009-1.38) | 0.08 |
| | 13 −24 | 53(88.3) | 7(11.7) | 1.82 (0.71-4.63) | 0.208 | 1.06 (0.19-5.88) | 0.94 |
| | > 24 | 79(80.6) | 19(19.4) | 1 | | 1 | |
| Whether mother breastfeeds when she is sick | Yes | 106(79.1) | 28(20.9) | 1 | | 1 | |
| | No | 40(87.0) | 6(13.0) | 1.76 (0.67-4.57) | 0.24 | 1.78 (0.34-9.12) | 0.48 |
| Whether mother breastfeeds when child is sick | Yes | 110(80.3) | 27(19.7) | 1 | | – | |
| | No | 36(83.7) | 7(16.3) | 1.26 (0.507-3.14) | 0.61 | – | |
| Whether washes hand before feeding the child | Yes | 94(79.7) | 24(20.3) | 1 | | – | |
| | No | 9(90.0) | 1(10.0) | 2.29 (0.27-19.03) | 0.44 | – | |
| | Sometimes | 43(82.7) | 9(17.3) | 1.22 (0.52-2.84) | 0.64 | – | |
| Washes fruits and vegetables | Yes | 143(81.2) | 33(18.8) | 1 | | – | |
| | No | 3(75.0) | 1(25.0) | 0.69 (0.07-6.86) | 0.75 | – | |
| Freshly prepares meals | Yes | 98(80.3) | 24(19.7) | 1 | | – | |
| | No | 48(82.8) | 10(17.2) | 1.17 (0.52-2.65) | 0.69 | – | |
| Supplementary nutrition utilized by child | Yes | 123(80.4) | 30(19.6) | 1 | | – | |
| | No | 20(87.0) | 3(13.0) | 1.62 (0.45-5.83) | 0.45 | – | |
| | Not sure | 3(75.0) | 1(25.0) | 0.73 (0.07-7.28) | 0.79 | – | |
| Whether house is visited by accredited social health activist (ASHA)/ Community health officer (CHO)[a] | Yes | 136(82.4) | 29(17.6) | 1 | | 1 | |
| | No | 10(66.7) | 5(33.3) | 0.42 (0.13-1.34) | 0.14 | 0.08 (0.01-0.77) | 0.02* |
| Presence of nearby health facility | Yes | 143(81.7) | 32(18.3) | 1 | | 1 | |
| | No | 3(60.0) | 2(40.0) | 0.33 (0.05-2.09) | 0.24 | 0.11 (0.000108-124.92) | 0.11 |

Note: Nine children were not tested for anaemia since they were below one year, 10 children refused haemoglobin test.

(CHO)[a] – grassroot level workers who are responsible for house visit at the subcentre or primary health centre level.

*statistically significant.

while Sunny R et al. also reported a higher percentage of PVTG children (Paniya, Bett Kurumba, Mula Kurumba, and Katunayaka) being malnourished in Nilgiris of Tamil Nadu (63%-underweight; 62%-stunting; 31%-wasting) [26]. Additionally, a study by Narayanappa D et al. on Jenukuruba (PVTG) tribal children in Mysore district found that 62.9% of children under six years of age were malnourished [27]. Similarly, Jaiswal A et al. studied the Irular tribe children in Villupuram, Tamil Nadu, and found that 58.43% were underweight, 64.04% were stunted, and 51.69% were wasted [28]. Although,

**Table 5. Association between wasting and sociodemographic, maternal and child characteristics (n = 180).**

| Characteristic | Category | Wasting | | COR with 95%CI | p value | AOR with 95%CI | p value |
|---|---|---|---|---|---|---|---|
| | | **Present (n = 141) n (%)** | **Absent (n = 39) n (%)** | | | | |
| **Sociodemographic characteristics** | | | | | | | |
| Type of family | Nuclear | 39(81.2) | 9(18.8) | 1 | | | |
| | Joint | 86(76.8) | 26(23.2) | 0.76 (0.32-1.78) | 0.53 | | |
| | Third generation | 16(80.0) | 4(20.0) | 0.92 (0.24-3.43) | 0.905 | | |
| Number of Family members | ≤5 members | 54(79.4) | 14(20.6) | 1 | | | |
| | >6 members | 87(77.7) | 25(22.3) | 0.902 (0.43-1.88) | 0.78 | | |
| Socioeconomic status | Middle | 94(77.0) | 28(23.0) | 1 | | | |
| | Lower | 47(81.0) | 11(19.0) | 1.27 (0.58-2.77) | 0.54 | | |
| Type of ration card | APL | 7(63.6) | 4(36.4) | 1 | | 1 | |
| | BPL/ Antyodaya/No ration card | 134(79.3) | 35(20.7) | 2.18 (0.606-7.89) | **0.23** | 4.47 (0.82-24.15) | 0.08 |
| Type of house | Kutcha House | 24(70.6) | 10(29.4) | 0.43 (0.08-2.33) | 0.33 | | |
| | Semi-Pucca house | 106(79.7) | 27(20.3) | 0.71 (0.14-3.41) | 0.67 | | |
| | Pucca house | 11(84.6) | 2(15.4) | 1 | | | |
| Type of cooking fuel | Wood stove | 89(78.8) | 24(21.2) | 1.07 (0.51-2.22) | 0.85 | | |
| | LPG stove | 52(77.6) | 15(22.4) | 1 | | | |
| Source of drinking water | Open well | 77(80.2) | 19(19.8) | 1.26 (0.62-2.57) | 0.51 | | |
| | Bore well | 64(76.2) | 20(23.8) | 1 | | | |
| Drinking water treatment before use | Yes | 104(75.4) | 34(24.6) | 1 | | 1 | |
| | No | 37(88.1) | 5(11.9) | 2.41 (0.88-6.64) | 0.08 | 2.37 (0.62-9.01) | 0.204 |
| Presence of household toilet | Yes | 131(78.4) | 36(21.6) | 1 | | | |
| | No | 10(76.9) | 3(23.1) | 0.91 (0.23-3.505) | 0.89 | | |
| **Maternal characteristics** | | | | | | | |
| Educational status of mother | Illiterate or upto 4 years of schooling | 18(72.0) | 7(28.0) | 1.02 (0.28-3.72) | 0.96 | | |
| | 5-10 years of schooling | 108(80.6) | 26(19.4) | 1.66 (0.58-4.69) | 0.33 | | |
| | >10 years of schooling | 15(71.4) | 6(28.6) | 1 | | | |
| Education of spouse | Illiterate or upto 4 years of schooling | 48(78.7) | 13(21.3) | 1.23 (0.406-3.73) | 0.71 | | |
| | 5-10 years of schooling | 75(78.9) | 20(21.1) | 1.25 (0.43-3.56) | 0.67 | | |
| | >10 years of schooling | 18(75.0) | 6(25.0) | 1 | | | |
| Age at marriage (in years) | <18 | 14(73.7) | 5(26.3) | 0.75(0.25-2.22) | 0.604 | | |
| | ≥18 | 127(78.9) | 34(21.1) | 1 | | | |

*(Continued)*

**Table 5.** (Continued)

| Characteristic | Category | Wasting | | COR with 95%CI | p value | AOR with 95%CI | p value |
|---|---|---|---|---|---|---|---|
| | | Present (n=141) n (%) | Absent (n=39) n (%) | | | | |
| Age at first conception (in years) | ≤30 | 122(77.2) | 36(22.8) | 0.53(0.15-1.91) | 0.33 | | |
| | >30 | 19(86.4) | 3(13.6) | 1 | | | |
| Number of times conceived | Once | 70(77.8) | 20(22.2) | 1 | | | |
| | ≥ Two | 71(78.9) | 19921.1) | 1.06 (0.52-2.17) | 0.85 | | |
| Number of ANC check ups | <Four times | 8(80.0) | 2(20.0) | 1.11 (0.22-5.46) | 0.89 | | |
| | ≥Four times | 133(78.2) | 37(21.8) | 1 | | | |
| Place of delivery | Hospital | 137(78.3) | 38(21.7) | 1 | | | |
| | Home | 4(80.0) | 1(20.0) | 1.109 (0.12-10.22) | 0.92 | | |
| Type of delivery | Normal | 86(86.9) | 13(13.1) | 1 | | | |
| | Cesarean section | 73(90.1) | 8(9.9) | 1.07 (0.52-2.19) | 0.84 | | |
| Calorie intake among women | Inadequate | 139(78.1) | 39(21.9) | – | | | |
| Protein intake among women | Adequate | 7(87.5) | 1(12.5) | 1 | | | |
| | Inadequate | 132(77.6) | 38(22.4) | 0.49 (0.05-4.16) | 0.51 | | |
| Calcium intake among women | Inadequate | 139(78.1) | 39(21.9) | – | | | |
| Iron intake among women | Adequate | 3(100.0) | 0(0) | – | | | |
| | Inadequate | 136(77.7) | 39(22.3) | – | | | |
| Retinol intake among women | Adequate | 15(75.0) | 5(25.0) | 1 | | | |
| | Inadequate | 124(78.5) | 34(21.5) | 1.21 (0.41-3.58) | 0.72 | | |
| BMI of mother | Under weight | 67(82.7) | 14(17.3) | 1.46 (0.66-3.18) | 0.34 | | |
| | Normal weight | 59(76.6) | 18(23.4) | 1 | | | |
| | Overweight & Obesity | 15(68.2) | 7(31.8) | 0.65 (0.23-1.85) | 0.42 | | |
| Hemoglobin level of mother | Normal | 74(74.0) | 26(26.0) | 1 | | 1 | |
| | Mild anaemia | 31(75.6) | 10(24.4) | 1.08 (0.47-2.52) | 0.84 | 0.78 (0.28-2.12) | 0. |
| | Moderate anaemia | 13(92.9) | 1(7.1) | 4.56 (0.56-36.65) | 0.15 | 9.43 (0.78-113.65) | 0.07 |
| | Severe anaemia | 21(91.3) | 2(8.7) | 3.68 (0.809-16.82) | 0.092 | 4.27 (0.71-25.51) | 0.11 |
| **Characteristics of children** | | | | | | | |
| Age (in months) | Below 24 months | 35(60.3) | 23(39.7) | 1 | | 1 | |
| | 25-60 months | 106(86.9) | 16(13.1) | 4.35 (2.06-9.15) | <0.001 | 5.05 (2.13-11.99) | <0.001* |
| Sex | Male | 68(79.1) | 18(20.9) | 1 | | | |
| | Female | 73(77.7) | 21(22.3) | 0.92 (0.45-1.87) | 0.81 | | |

*(Continued)*

**Table 5.** (Continued)

| Characteristic | Category | Wasting | | COR with 95%CI | p value | AOR with 95%CI | p value |
|---|---|---|---|---|---|---|---|
| | | Present (n=141) n (%) | Absent (n=39) n (%) | | | | |
| Hospital admission in neonatal period | Yes | 14(66.7) | 7(33.3) | 0.504 (0.18-1.35) | **0.17** | 0.506 (0.15-1.66) | 0.26 |
| | No | 127(79.9) | 32(20.1) | 1 | | 1 | |
| Birth weight | Low birth weight | 58(80.6) | 14(19.4) | 1.26 (0.06-2.63) | 0.53 | | |
| | Normal weight | 82(76.6) | 25(23.4) | 1 | | | |
| Immunization status | Fully immunized | 137(77.8) | 39(22.2) | – | | | |
| | Partially immunized | 4(100.0) | 0(0) | – | | | |
| Positive history of Measles in last six months | Yes | 8(88.9) | 1(11.1) | 2.28 (0.27-18.85) | 0.44 | | |
| | No | 133(77.8) | 38(22.2) | 1 | | | |
| History of ARI/ADD in last six months | Yes | 6(54.5) | 5(45.5) | 0.302 (0.08-1.05) | 0.06 | 0.22 (0.04-1.17) | 0.07 |
| | No | 135(79.9) | 34(20.1) | 1 | | 1 | |
| Worm infestation in last six months | Yes | 14(93.3) | 1(6.7) | 4.18 (0.53-32.89) | 0.17 | 2.98 (0.28-30.84) | 0.36 |
| | No | 127(77.0) | 38(23.0) | 1 | | 1 | |
| History of Ingestion of clay/soil | Yes | 15(83.3) | 3(16.7) | 1.42 (0.39-5.209) | 0.58 | | |
| | No | 126(77.8) | 36(22.2) | 1 | | | |
| Hospitalization of child in last one year for acute illness | Yes | 25(64.9) | 11(30.6) | 0.54 (0.24-1.24) | 0.15 | 0.83 (0.28-2.39) | 0.73 |
| | No | 116(80.6) | 28(19.4) | 1 | | 1 | |
| Calorie intake | Adequate | 27(79.4) | 7(20.6) | 1 | | | |
| | Inadequate | 114(78.1) | 32(21.9) | 0.92(0.36-2.31) | 0.86 | | |
| Protein intake | Adequate | 99(83.2) | 20(16.8) | 1 | | | |
| | Inadequate | 42(68.9) | 19(31.1) | 0.44 (0.21-0.92) | 0.02 | 0.27 (0.11-0.69) | 0.006* |
| Calcium intake | Adequate | 6(85.7) | 1(14.3) | 1 | | 1 | |
| | Inadequate | 135(78.0) | 38(22.0) | 0.59 (0.06-5.07) | 0.63 | | |
| Iron intake | Adequate | 27(73.0) | 10(27.0) | 1 | | | |
| | Inadequate | 114(79.7) | 29(20.3) | 1.45 (0.63-3.34) | 0.37 | | |
| Retinol intake | Adequate | 54(76.1) | 17(23.9) | 1 | | | |
| | Inadequate | 87(79.8) | 22(20.2) | 1.24 (0.607-2.55) | 0.55 | | |
| Breast feeding initiation | Immediately after birth | 82(76.6) | 25(23.4) | 1 | | | |
| | Half an hour-4 hours | 56(82.4) | 12(17.6) | 1.42 (0.66-3.06) | 0.36 | | |
| | Beyond 4hrs | 3(60.0) | 2(40.0) | 0.45 (0.07-2.89) | 0.406 | | |
| Pre lacteal feeding to infant | Yes | 17(81.0) | 4(19.0) | 1.200 (0.37-3.79) | 0.75 | | |
| | No | 124(78.0) | 35(22.0) | 1 | | | |

*(Continued)*

**Table 5.** (Continued)

| Characteristic | Category | Wasting | | COR with 95%CI | p value | AOR with 95%CI | p value |
|---|---|---|---|---|---|---|---|
| | | Present (n = 141) n (%) | Absent (n = 39) n (%) | | | | |
| Duration of breast feeding to child (months) | Upto 6 months | 8(80.0) | 2(20.0) | 0.96 (0.18-4.902) | 0.96 | | |
| | 7 −12 months | 9(75.0) | 3(25.0) | 0.72(0.17-2.92) | 0.64 | | |
| | 13 −24 months | 54(90.0) | 6(10.0) | 0.72 (0.33-1.55) | 0.406 | | |
| | > 24 months | 88(89.8) | 10(10.2) | 1 | | | |
| Whether mother breastfeeds when she is sick | Yes | 119(88.9) | 15(11.2) | 1 | | | |
| | No | 40(87.0) | 6(13.0) | 0.99 (0.44-2.24) | 0.98 | | |
| Whether mother breastfeeds when child is sick | Yes | 121(88.3) | 16(11.7) | 1.56 (0.63-3.86) | 0.32 | | |
| | No | 38(88.4) | 5(11.6) | 1 | | | |
| Whether washes hand before feeding the child | Yes | 89(75.4) | 29(24.6) | 1 | | 1 | |
| | No | 8(80.0) | 2(20.0) | 1.303 (0.26-6.48) | 0.74 | 0.97 (0.13-6.92) | |
| | Sometimes | 44(84.6) | 8(15.4) | 1.79 (0.75-4.24) | 0.18 | 0.98 (0.32-3.006) | 0.98 |
| Washes fruits and vegetables | Yes | 138(78.4) | 38(21.6) | 1 | | | |
| | No | 3(75.0) | 1(25.0) | 0.82 (0.08-8.17) | 0.87 | | |
| Freshly prepares meals | Yes | 93(76.2) | 29(23.8) | 1 | | | |
| | No | 48(82.8) | 10(17.2) | 1.49 (0.67-3.32) | 0.32 | | |
| Supplementary nutrition utilized by child | Yes | 119(77.8) | 34(22.2) | 1 | | | |
| | No | 19(82.6) | 4(17.4) | 1.35 (0.43-4.25) | 0.601 | | |
| | Not sure | 3(75.0) | 1(25.0) | 0.85 (0.08-8.507 | 0.89 | | |
| Whether house is visited by accredited social health activist (ASHA)/ Community health officer (CHO)[a] | Yes | 128(77.6) | 37(22.4) | 1 | | | |
| | No | 13(86.7) | 2(13.3) | 1.87 (0.406-8.703) | 0.42 | | |
| Presence of nearby health facility | Yes | 137(78.3) | 38(21.7) | 1 | | | |
| | No | 4(80.0) | 1(20.0) | 1.109 (0.12-10.22) | 0.92 | | |

Note: Nine children were not tested for anaemia since they were below one year, 10 children refused Hemoglobin test.

(CHO)a – grassroot level workers who are responsible for house visit at the subcenter or primary health center level, *statistically significant.

these studies have documented high levels of malnutrition among PVTG children, the Koraga children in our study exhibited the highest prevalence.

Further, studies among non-PVTG tribal children have also reported higher prevalence of malnutrition; however, the rates are much below the present study findings. A study on Kadukuruba tribal children from Mysore district by Manjunath R et al. reported high malnutrition rates, with 60.4% being underweight, 55.4% with stunting, and 43% with wasting

**Table 6. Association between sociodemographic, maternal and child characteristics and underweight n = 180.**

| Characteristic | Category | Underweight | | COR with 95%CI | p value | AOR with 95%CI | p value |
|---|---|---|---|---|---|---|---|
| | | Present (n = 159) n(%) | Absent (n = 21) n(%) | | | | |
| **Sociodemographic characteristics** | | | | | | | |
| Type of family | Nuclear | 40(83.3) | 8(16.7) | 0.26 (0.03-2.25) | 0.22 | 0.109 (0.006-1.94) | 0.13 |
| | Joint | 100(89.3) | 12(10.7) | 0.43 (0.05-3.57) | 0.44 | 0.15(0.01-2.29) | 0.17 |
| | Third generation | 19(95.0) | 1(5.0) | 1 | | 1 | |
| Number of Family members | ≤5 members | 57(83.8) | 11(16.2) | 1 | | 1 | |
| | >6 members | 102(91.1) | 10(8.9) | 1.96 (0.78-4.91) | 0.14 | 1.49 (0.37-6.03) | 0.57 |
| Socioeconomic status | Middle | 105(86.1) | 17(13.9) | 1 | | 1 | |
| | Lower | 54(93.1) | 4(6.9) | 2.18 (0.701-6.81) | 0.17 | 2.83 (0.66-12.12) | 0.16 |
| Type of ration card | APL | 8(72.7) | 3(27.3) | 1 | | 1 | |
| | BPL/ Antyodaya/No ration card | 151(89.3) | 18(10.7) | 3.14 (0.76-12.93) | 0.11 | 1.32 (0.19-9.13) | 0.77 |
| Type of house | Kutcha House | 31(91.2) | 3(8.8) | 0.86(0.08-9.11) | 0.901 | | |
| | Semi-Pucca house | 116(87.2) | 17(12.8) | 0.56 (0.06-4.65) | 0.56 | | |
| | Pucca house | 12(92.3) | 1(7.7) | 1 | | | |
| Type of cooking fuel | Wood stove | 98(86.7) | 15(13.3) | 0.64 (0.23-1.74) | 0.38 | | |
| | LPG stove | 61(91.0) | 6(9.0) | 1 | | | |
| Source of drinking water | Open well | 86(89.6) | 10(10.4) | 1.29 (0.52-3.22) | 0.57 | | |
| | Bore well | 73(86.9) | 11(13.1) | 1 | | | |
| Drinking water treatment before use | Yes | 120(87.0) | 18(13.0) | 1 | | | |
| | No | 39(92.9) | 3(7.1) | 1.95 (0.54-6.97) | 0.304 | | |
| Presence of household toilet | Yes | 150(89.8) | 17(10.2) | 1 | | 1 | |
| | No | 9(69.2) | 4(30.8) | 0.25 (0.07-0.91) | 0.03 | 0.08 (0.01-0.48) | 0.006* |
| **Maternal characteristics** | | | | | | | |
| Educational status of mother | Illiterate or upto 4 years of schooling | 21(84.0) | 4(16.0) | 0.87 (0.17-4.43) | 0.87 | | |
| | 5-10 years of schooling | 120(89.6) | 14(10.4) | 1.42 (0.37-5.46) | 0.602 | | |
| | >10 years of schooling | 18(85.7) | 3(14.3) | 1 | | | |
| Education of spouse | Illiterate or upto 4 years of schooling | 53(86.9) | 8(13.1) | 0.28 (0.03-2.43) | 0.25 | | |
| | 5-10 years of schooling | 83(87.4) | 12(12.6) | 0.301 (0.03-2.43) | 0.26 | | |
| | >10 years of schooling | 23(95.8) | 1(4.2) | 1 | | | |
| Age at marriage (in years) | <18 | 16(84.2) | 3(15.8) | 0.67 (0.17-2.53) | 0.55 | | |
| | ≥18 | 143(88.8) | 18(11.2) | 1 | | | |

*(Continued)*

| Characteristic | Category | Underweight | | COR with 95%CI | p value | AOR with 95%CI | p value |
|---|---|---|---|---|---|---|---|
| | | Present (n=159) n(%) | Absent (n=21) n(%) | | | | |
| Age at first conception (in years) | ≤30 | 142(89.9) | 16(10.1) | 2.61 (0.84-8.02) | 0.09 | 4.47 (0.98-20.26) | 0.05 |
| | >30 | 17(77.3) | 5(22.7) | 1 | | 1 | |
| Number of times conceived | Once | 81(90.0) | 9(10.0) | 1 | | | |
| | ≥ Two | 78(86.7) | 12(13.3) | 0.72 (0.28-1.81) | 0.48 | | |
| Number of ANC check ups | <Four times | 9(90.0) | 1(10.0) | 1.20 (0.14-9.97) | 0.86 | | |
| | ≥Four times | 150(88.2) | 20(11.8) | 1 | | | |
| Place of delivery | Hospital | 155(88.6) | 20(11.4) | 1 | | | |
| | Home | 4(80.0) | 1(20.0) | 0.51 (0.05-4.84) | 0.56 | | |
| Type of delivery | Normal | 86(86.9) | 13(13.1) | 1 | | | |
| | Cesarean section | 73(90.1) | 8(9.9) | 1.37 (0.54-3.51) | 0.500 | | |
| Calorie intake among women | Inadequate | 157(88.2) | 21(11.8) | – | | | |
| Protein intake among women | Adequate | 7(85.7) | 1(12.5) | 1 | | | |
| | Inadequate | 150(88.2) | 20(11.8) | 1.07 (0.12-9.16) | 0.95 | | |
| Calcium intake among women | Inadequate | 157(88.2) | 21(11.8) | – | | | |
| Iron intake among women | Adequate | 2(66.7) | 1(33.3) | 1 | | | |
| | Inadequate | 155(88.6) | 20(11.4) | 3.87 (0.33-44.69) | 0.27 | | |
| Retinol intake among women | Adequate | 18(90.0) | 2(10.0) | 1 | | | |
| | Inadequate | 139(88.0) | 19(12.0) | 0.81 (0.17-3.78) | 0.79 | | |
| BMI of mother | Under weight | 71(87.7) | 10(12.3) | 0.82 (0.37-2.209) | 0.69 | | |
| | Normal weight | 69(89.6) | 8(10.4) | 1 | | | |
| | Overweight & Obesity | 19(86.4) | 3(13.6) | 0.73 (0.17-3.04) | 0.67 | | |
| Hemoglobin level of mother | Normal | 91(91.0) | 9(9.0) | | | | |
| | Mild anaemia | 33(80.5) | 8(19.5) | | | | |
| | Moderate anaemia | 14(100.0) | 0(0) | – | | | |
| | Severe anaemia | 19(82.6) | 4(17.4) | | | | |
| **Characteristics of children** | | | | | | | |
| Age (in months) | Below 24 months | 48(82.8) | 10(17.2) | 1 | | 1 | |
| | 25-60 months | 111(91.0) | 11(9.0) | 2.10 (0.83-5.28) | 0.11 | 3.08 (0.86-11.06) | 0.08 |
| Sex | Male | 76(88.4) | 10(11.6) | 1 | | | |
| | Female | 83(88.3) | 11(11.7) | 0.99 (0.39-2.46) | 0.98 | | |
| Hospital admission in neonatal period | Yes | 18(85.7) | 3(14.3) | 0.76 (0.205-2.85) | 0.69 | | |
| | No | 141(88.7) | 18(11.3) | 1 | | | |

*(Continued)*

**Table 6.** (Continued)

| Characteristic | Category | Underweight | | COR with 95%CI | p value | AOR with 95%CI | p value |
|---|---|---|---|---|---|---|---|
| | | Present (n=159) n(%) | Absent (n=21) n(%) | | | | |
| Birth weight | Low birth weight | 65(90.3) | 7(9.7) | 1.39 (0.53-3.65) | 0.49 | | |
| | Normal weight | 93(86.9) | 14(13.1) | 1 | | | |
| Immunization status | Fully immunized | 155(88.1) | 21(11.9) | – | | | |
| | Partially immunized | 4(100.0) | 0(0) | – | | | |
| Positive history of Measles in last six months | Yes | 8(88.9) | 1(11.1) | 1.06 (0.12-8.92) | 0.95 | | |
| | No | 151(88.3) | 20(11.7) | 1 | | | |
| History of ARI/ADD in last six months | Yes | 11(100.0) | 0(0) | – | | | |
| | No | 148(87.6) | 21(12.4) | – | | | |
| Worm infestation in last six months | Yes | 14(93.3) | 1(6.7) | 1.93(0.24-15.48) | 0.53 | | |
| | No | 145(87.9) | 20(12.1) | 1 | | | |
| History of Ingestion of clay/soil | Yes | 17(94.4) | 1(5.6) | 2.39 (0.302-18.98) | 0.408 | | |
| | No | 142(87.7) | 20(12.3) | 1 | | | |
| Hospitalization of child in last one year for acute illness | Yes | 34(94.4) | 2(5.6) | 2.58 (0.57-11.64) | 0.21 | 4.68 (0.65-33.68) | 0.12 |
| | No | 125(86.8) | 19(13.2) | 1 | | 1 | |
| Calorie intake | Adequate | 28(82.4) | 6(17.6) | 1 | | 1 | |
| | Inadequate | 131(89.7) | 15(10.3) | 1.87 (0.66-5.24) | 0.23 | 0.73 (0.14-3.64) | 0.708 |
| Protein intake | Adequate | 105(88.2) | 14(11.8) | 1 | | | |
| | Inadequate | 54(88.5) | 7(11.5) | 1.02(0.39-2.69) | 0.95 | | |
| Calcium intake | Adequate | 6(85.7) | 1(14.3) | 1 | | | |
| | Inadequate | 153(88.4) | 20(11.6) | 1.27(0.14-11.14) | 0.82 | | |
| Iron intake | Adequate | 30(81.1) | 7(18.9) | 1 | | 1 | |
| | Inadequate | 129(90.2) | 14(9.8) | 2.15 (0.79-5.78)3 | 0.13 | 2.82 (0.68-11.55) | 0.14 |
| Retinol intake | Adequate | 61(85.9) | 10(14.1) | 1 | | | |
| | Inadequate | 98(89.9) | 11(10.1) | 1.46 (0.58-3.64) | 0.41 | | |
| Breast feeding initiation | Immediately after birth | 97(90.7) | 10(9.3) | 1 | | | |
| | Half an hour-4 hours | 58(85.3) | 10(14.7) | 0.59 (0.23-1.52) | 0.28 | | |
| | Beyond 4hrs | 4(80.0) | 1(20.0) | 0.41 (0.04-4.05) | 0.44 | | |
| Pre lacteal feeding to infant | Yes | 18(85.7) | 3(14.3) | 0.76 (0.205-2.85) | 0.69 | | |
| | No | 141(88.7) | 18(11.3) | 1 | | | |

*(Continued)*

**Table 6.** (Continued)

| Characteristic | Category | Underweight | | COR with 95%CI | p value | AOR with 95%CI | p value |
|---|---|---|---|---|---|---|---|
| | | Present (n=159) n(%) | Absent (n=21) n(%) | | | | |
| Duration of breast feeding to child (months) | Upto 6 months | 8(80.0) | 2(20.0) | 0.45 (0.08-2.44) | 0.35 | 0.52 (0.07-3.81) | 0.52 |
| | 7−12 months | 9(75.0) | 3(25.0) | 0.34 (0.07-1.47) | 0.14 | 0.49 (0.06-4.03) | 0.51 |
| | 13−24 months | 54(90.0) | 6(10.0) | 1.02 (0.35-2.97) | 0.96 | 1.62 (0.43-6.09) | 0.47 |
| | > 24 months | 88(89.8) | 10(10.2) | 1 | | 1 | |
| Whether mother breastfeeds when she is sick | Yes | 119(88.8) | 15(11.2) | 1 | | | |
| | No | 40(87.0) | 6(13.0) | 0.84 (0.305-2.31) | 0.73 | | |
| Whether mother breastfeeds when child is sick | Yes | 121(88.3) | 16(11.7) | 1 | | | |
| | No | 38(88.4) | 5(11.6) | 1.005 (0.34-2.92) | 0.99 | | |
| Whether washes hand before feeding the child | Yes | 104(88.1) | 14(11.9) | 1 | | 1 | |
| | No | 7(70.0) | 3(30.0) | 0.31 (0.07-1.35) | 0.12 | 0.47 (0.06-3.45) | 0.46 |
| | Sometimes | 48(92.3) | 4(7.7) | 1.61 (0.505-5.16) | 0.41 | 3.71 (0.66-20.79) | 0.13 |
| Washes fruits and vegetables | Yes | 156(88.6) | 20(11.4) | 1 | | | |
| | No | 3(75.0) | 1(25.0) | 0.38 (0.03-3.87) | | | |
| Freshly prepares meals | Yes | 111(91.0) | 11(9.0) | 1 | | 1 | |
| | No | 48(82.8) | 10(17.2) | 0.47 (0.18-1.19) | 0.11 | 0.24 (0.07-0.89) | 0.03* |
| Supplementary nutrition utilized by child ICDS | Yes | 134(87.6) | 19(12.4) | 1 | | | |
| | No | 21(91.3) | 2(8.7) | – | | | |
| | Not sure | 4(100.0) | 0(0) | – | | | |
| Whether house is visited by accredited social health activist (ASHA)/ Community health officer (CHO)[a] | Yes | 146(88.5) | 19(11.5) | 1 | | | |
| | No | 13(86.7) | 2(13.3) | 0.84 (0.17-4.04) | 0.83 | | |
| Presence of nearby health facility | Yes | 155(88.6) | 20(11.4) | 1 | | | |
| | No | 4(80.0) | 1(20.0) | 0.51 (0.05-4.84) | 0.56 | | |

Note: Nine children were not tested for anaemia since they were below one year, 10 children refused Hemoglobin test.

(CHO)[a] – grassroot level workers who are responsible for house visit at the subcenter or primary health center level *statistically significant.

[29]. Similarly, in Wayanad, Kerala, a study by Philip RR et al. conducted among tribal preschool children reported a high prevalence of malnutrition (39%-underweight; 38%-stunting; 20.5%-wasting) [30]. Likewise, another study from southern India, by Reddy VB et al. on Sugali tribal children in Chittoor district, Andhra Pradesh, also reported similar findings (32.7%- underweight;18.3%-wasting; 38.3%-stunting) [31]. However, a study by Duwarah S et al. in a tertiary care hospital in Shillong, Meghalaya, reported a prevalence of underweight at 19.7%, stunting at 35.5%, and wasting at 8.5% among preschool tribal children, which is significantly lower compared to the prevalence observed in our study population [32]. These differences may be attributed to variations in geographical locations and study settings.

In our study, stunting was associated with drinking water from an open well, ≥ two conceptions, low birth weight, poor retinol intake, lack of exclusive breastfeeding, and lack of visits by ASHA, while wasting was linked to older children and inadequate protein intake. A study by Adhikari T et al. found that stunting, wasting, and underweight among tribal children were associated with factors like age, maternal BMI, education, and household wealth. Like the study by Adhikari T et al., which reported a 65.7% prevalence of wasting among children under two years of age, our findings also indicate a similar prevalence (60.3%) in this age group [33].

Our study identified that children aged 25–60 months were having a higher likelihood of wasting, aligning with findings from a study by Meshram II et al. conducted in Integrated Tribal Development Agency (ITDA) areas of nine states across South, East, West, and Central India among tribal children. The study found that children aged 1–3 years were significantly associated with underweight and stunting, while those aged 3–5 years were associated with all three indicators of malnutrition (underweight, stunting, and wasting) [34]. In contrast, a study by Ghosh S et al. among tribal children of Warli, Malhar Koli tribes and Katkari a PVTG of Palghar district, Maharashtra, found no association between age 2–5 years and wasting, while age 3–5 years was linked to underweight, and age 4–5 years had a lower likelihood of stunting [35].

In the present study, low birth weight was a significant predictor of stunting. However, underweight and wasting did not show a significant association. This supports with a study by Singh A et al. among Khasi tribes of Meghalaya, which also found that low birth weight was significantly associated with underweight and stunting [36]. Similarly, a study by Senthilkumar SK et al. in a tribal community of Coimbatore district, which considered malnutrition as the presence of any one of the three parameters consisting of underweight, wasting, or stunting, found that low birth weight babies had a higher risk of malnutrition [37].

Our study did not find any association between the gender of the children, the educational status of parents and indicators of malnutrition. Unlike our findings, study by Sunny R et al. in the tribal population of the Nilgiris, Tamil Nadu, found that male children, age > 2 years, and paternal illiteracy were significantly associated with being underweight, while age > 2 years and maternal illiteracy were associated with stunting and male children and having low socioeconomic status were associated with the wasting [25].

Reducing malnutrition among tribal children needs a comprehensive, multifaceted approach that integrates socio-cultural, environmental, and healthcare considerations. Breaking this vicious cycle requires not only improved nutrition but also enhanced healthcare access, socio-economic support, and targeted health education.

A key strength of this study lies in its comprehensive approach, assessing sociodemographic, maternal, child-related, clinical, and environmental factors. However, the present study also has certain limitations. The study relied on self-reported data for some factors, such as breastfeeding practices and morbidity, which may have led to recall bias. The study used the 24-hour dietary recall method, which may not reflect long-term dietary intake, a detailed dietary analysis is needed for a comprehensive evaluation. The study did not assess specific cultural factors such as dietary practices, food taboos, and traditional cooking methods which could have affected nutritional status of under-five children and should be considered in future research. Few associations observed across stunting, wasting and underweight in the present study were not consistent and showed inverse associations with available biological and epidemiological evidence. These findings may be influenced by the high prevalence of malnutrition and small subgroup sizes in the population studied which could have affected the estimates. Therefore, these associations should be interpreted with caution. Additionally, as the study focused exclusively on under five children from Koraga PVTG community, the findings may not be generalized to other tribal groups.

This study highlights the severe malnutrition burden among Koraga tribal children, with a high prevalence of underweight, stunting, and wasting. Findings of the present study further emphasizes the complex interplay of environmental, dietary, and maternal health factors which contribute to malnutrition. To address the high burden of malnutrition among Koraga tribal children, efforts should focus on strengthening maternal and child healthcare services through improving maternal nutrition, ensuring proper antenatal care, addressing high-risk pregnancies and enhancing ASHA/CHO home

visits to prevent low birth weight, which can help reduce malnutrition. Further, continuous growth monitoring, a nutritious diet, better sanitation, access to clean water, and timely healthcare interventions can improve child health outcomes. Policy interventions should ensure targeted nutrition programs under ICDS and Prime Minister's Overarching Scheme for Holistic Nutrition (POSHAN) Abhiyaan, along with community-led monitoring to improve the effectiveness of existing interventions.

## Supporting information

**S1 File. Inclusivity in global research.**
(DOCX)

## Acknowledgments

The Authors acknowledge Integrated Tribal Development Project, Udupi district; the Department of Health and Family Welfare, Udupi district, and Samagra Grameena Ashrama, Udupi for providing administrative and technical support for carrying out this study.

## Author contributions

**Conceptualization:** Sneha Deepak Mallya, Biju Soman, Dinesh M Nayak, Ranjitha S Shetty.

**Data curation:** Biju Soman, Srikanth Gouspure.

**Formal analysis:** Sneha Deepak Mallya, Biju Soman, Ranjitha S Shetty.

**Funding acquisition:** Ranjitha S Shetty.

**Investigation:** Biju Soman, Srikanth Gouspure.

**Methodology:** Sneha Deepak Mallya, Dinesh M Nayak, Ashwini Kumar, Harpreet Kaur, Ranjitha S Shetty.

**Project administration:** Sneha Deepak Mallya, Dinesh M Nayak, Ashwini Kumar, Ranjitha S Shetty.

**Resources:** Ranjitha S Shetty.

**Software:** Biju Soman.

**Supervision:** Sneha Deepak Mallya, Dinesh M Nayak, Ashwini Kumar, Ranjitha S Shetty.

**Validation:** Sneha Deepak Mallya, Ranjitha S Shetty.

**Visualization:** Sneha Deepak Mallya, Harpreet Kaur.

**Writing – original draft:** Sneha Deepak Mallya, Biju Soman, Srikanth Gouspure, Pavan Kumar Thatoju, Ranjitha S Shetty.

**Writing – review & editing:** Sneha Deepak Mallya, Biju Soman, Srikanth Gouspure, Pavan Kumar Thatoju, Dinesh M Nayak, Ashwini Kumar, Harpreet Kaur, Ranjitha S Shetty.

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
