## [Decision Letter · Decision Letter 0]

28 Jan 2026

PONE-D-25-37335Nourishing futures: assessing nutritional health among children under-five years of age belonging to a particularly vulnerable tribal community in southern IndiaPLOS One

Dear Dr. Shetty,

Thank you for submitting your manuscript to PLOS ONE. After careful consideration, we feel that it has merit but does not fully meet PLOS ONE’s publication criteria as it currently stands. Therefore, we invite you to submit a revised version of the manuscript that addresses the points raised during the review process.

We look forward to receiving your revised manuscript.

Kind regards,

Biswajit Pal, M.SC., Ph.D

Academic Editor

PLOS One

Journal Requirements:

Reviewers' comments:

Reviewer's Responses to Questions

**Comments to the Author**

1. Is the manuscript technically sound, and do the data support the conclusions?

Reviewer #1: Yes

Reviewer #2: Yes

2. Has the statistical analysis been performed appropriately and rigorously? 

Reviewer #1: Yes

Reviewer #2: Yes

3. Have the authors made all data underlying the findings in their manuscript fully available?

Reviewer #1: Yes

Reviewer #2: Yes

4. Is the manuscript presented in an intelligible fashion and written in standard English?

Reviewer #1: Yes

Reviewer #2: Yes

5. Review Comments to the Author

Reviewer #1: The article gives information about the nutritional status of PVTG's children below 5 years of age. Most of the terms mentioned in the article are technical terms used by medical practitioners which may not be understood by the researchers of other fields. Explaining those terms in the footnotes would have made the readers understand the article better. Even though the statistical analysis was rigorous the mention of cultural factors was missing in the article which might also play a significant role in the determining the reasons for the stunned growth. Some of the associations concerning the malnutrition was ambiguous. Even though the article mentions about ITDS,ASHA etc their role concerning the malnutrition was not clear.

Reviewer #2: 1.Yes, it is a technically sound piece of scientific research with data that supports the conclusions.

2. Yes, the analysis has been performed appropriately.

3. Yes, It is.

4. The manuscript is written in a standard english.

The study addresses an important public health issue, as nutritional health among under five years children in Particularly Vulnerable Tribal Groups (PVTGs) remains a serious concern. The data appear to support the conclusions, and the statistical analyses have been conducted appropriately. However, several revisions are required to strengthen the scientific rigor and clarity of the manuscript.

The Introduction should include a brief description of the Koraga tribal community to provide adequate contextual background. The rationale for limiting the survey to 180 participants needs to be clearly explained. It should be specified whether this number represents the total population of under five children belonging to the Koraga community in the selected district or whether other factors determined the sample size.

In the Methodology section, the sampling procedure must be described in detail. The manuscript should clearly explain how the sample of 180 children was identified and recruited, including the underlying sampling framework and any inclusion or exclusion criteria. For infants and young children from birth to 24 months, it is essential to mention that, as per WHO recommendations, nutritional status was assessed using weight-for-length, along with length-for-age and weight-for-age indicators.

In the Results section, Table 2 reports data for 178 participants instead of 180. A clear explanation should be provided for this discrepancy, including whether any participants were excluded from the final analysis and the reasons for such exclusion. This information should be explicitly stated in the manuscript.

Finally, as the study focuses on under five years PVTG children, a dedicated section on study limitations is required.

Addressing these points through appropriate revisions is necessary to improve the scientific completeness and accuracy of the manuscript before it can be considered for publication.

6. PLOS authors have the option to publish the peer review history of their article (what does this mean?). If published, this will include your full peer review and any attached files.

Reviewer #1: **Yes:** Dr.G.V.Snigdha Raj

Reviewer #2: No

---

## [Author Response · Author response to Decision Letter 1]

6 Apr 2026

Dear Reviewers,

Thank you for taking the time to review our manuscript and for offering thoughtful, constructive comments. We sincerely appreciate your insights, which have helped us improve the clarity and overall quality of the work.

Warm regards

Dr Ranjitha S Shetty

---

## [Decision Letter · Decision Letter 1]

23 Apr 2026

Nourishing futures: assessing nutritional health among children under-five years of age belonging to a particularly vulnerable tribal community in southern India

PONE-D-25-37335R1

Dear Dr. Shetty,

We’re pleased to inform you that your manuscript has been judged scientifically suitable for publication and will be formally accepted for publication once it meets all outstanding technical requirements.

Kind regards,

Biswajit Pal, M.SC., Ph.D

Academic Editor

PLOS One

Additional Editor Comments (optional):

Reviewers' comments:

Reviewer's Responses to Questions

**Comments to the Author**

1. If the authors have adequately addressed your comments raised in a previous round of review and you feel that this manuscript is now acceptable for publication, you may indicate that here to bypass the “Comments to the Author” section, enter your conflict of interest statement in the “Confidential to Editor” section, and submit your "Accept" recommendation.

Reviewer #1: All comments have been addressed

Reviewer #2: All comments have been addressed

2. Is the manuscript technically sound, and do the data support the conclusions?

Reviewer #1: Yes

Reviewer #2: Yes

3. Has the statistical analysis been performed appropriately and rigorously? 

Reviewer #1: Yes

Reviewer #2: Yes

4. Have the authors made all data underlying the findings in their manuscript fully available?

Reviewer #1: Yes

Reviewer #2: Yes

5. Is the manuscript presented in an intelligible fashion and written in standard English?

Reviewer #1: Yes

Reviewer #2: Yes

6. Review Comments to the Author

Reviewer #1: The article is quantitatively sound and qualitative aspects would have added more strength to it. Apart from that its a good paper.

Reviewer #2: Authors have adequately addressed the comments raised in the previous round of review and this manuscript is now acceptable for publication. Now the manuscript is technically sound and appropriate for the scientific research with data that support conclusion.

7. PLOS authors have the option to publish the peer review history of their article (what does this mean?). If published, this will include your full peer review and any attached files.

Reviewer #1: No

Reviewer #2: No

---

## [Editor Report · Acceptance letter]

PONE-D-25-37335R1

PLOS One

Dear Dr. Shetty,

I'm pleased to inform you that your manuscript has been deemed suitable for publication in PLOS One. Congratulations! Your manuscript is now being handed over to our production team.

Kind regards,

on behalf of

Dr. Biswajit Pal

Academic Editor

PLOS One